# Evaluating a Novel Gas Sensor for Ambient Monitoring in Automated Life Science Laboratories

**DOI:** 10.3390/s22218161

**Published:** 2022-10-25

**Authors:** Mohammed Faeik Ruzaij Al-Okby, Thomas Roddelkopf, Heidi Fleischer, Kerstin Thurow

**Affiliations:** 1Technical Institute of Babylon, Al-Furat Al-Awsat Technical University (ATU), Kufa 54003, Iraq; 2Center for Life Science Automation (celisca), University of Rostock, 18119 Rostock, Germany; 3Institute of Automation, University of Rostock, 18119 Rostock, Germany

**Keywords:** gas sensors, hazardous gases, nitrogen oxide index (NOx-Index), indoor air quality (IAQ), internet of things (IoT), volatile organic compounds index (VOC-index)

## Abstract

Air pollution and leakages of hazardous and toxic gases and chemicals are among the dangers that frequently occur at automated chemical and life science laboratories. This type of accident needs to be processed as soon as possible to avoid the harmful side effects that can happen when a human is exposed. Nitrogen oxides and volatile organic compounds are among the most prominent indoor air pollutants, which greatly affect the lifestyles in these places. In this study, a commercial MOX gas sensor, SGP41, was embedded in an IoT environmental sensor node for hazardous gas detection and alarm. The sensor can detect several parameters, including nitrogen oxide index (NOx-Index) and volatile organic compound index (VOC-Index). Several tests were conducted to detect the leakage of nitrogen oxides and volatile organic compounds in different concentrations and volumes, as well as from different leakage distances, to measure the effect of these factors on the response speed and recovery time of the sensors used. These factors were also compared between the different sensors built into the sensor node to give a comprehensive picture of the system used. The system testing results revealed that the SGP41 sensor is capable of implementing the design purposes for the target parameters, can detect a small NO_2_ gas leakage starting from 0.3% volume, and can detect all the tested VOC solvents ≥ 100 µL

## 1. Introduction

Air pollution in workplaces, especially workplaces with possible sources of pollutants, is a common hazard that threatens the safety of workers and workflow. Nitrogen oxides and VOCs are among the main air pollutants that influence public safety in work environments. Statistics from the World Health Organization, as well as from the Global Burden of Disease epidemiological study, revealed that at least 7 million premature deaths are due to air pollutants in homes and the surrounding environment [1,2,3]. Across Europe, nitrogen oxide (NOx, mainly NO and NO_2_) and VOC pollution have become major environmental concerns. Some large urban areas already register higher nitrogen dioxide levels than EU safety limits. This situation is not bounded only to large industrial areas anymore. Many surrounding rural areas and small-sized populations frequently recorded violations of nitrogen oxide concentration [4,5,6,7]. The risks of exposure to nitrogen oxides are not limited to the possibility of acute pneumonia; they also include the possibility of permanent damage to the tissues and functions of the respiratory system caused by the expansion of the alveoli and the addition of airway volume, which may lead to the development of early focal emphysema [8,9].

Nowadays, semiconductor metal oxide sensors are commonly used due to some of their attractive properties such as low cost, small size, long life, ease of use, and their ability to detect a wide range of chemical gases and vapors. In addition, they are characterized by the ease of adjusting their output, which is usually in the form of a change in the electrical impedance of the sensing material [10,11,12,13]. 

Numerous products and research efforts have been implemented to detect and report excessive levels of NOx and VOC pollutants. Dong et al. [14] proposed a cloud-connected NO_2_ and ozone electrochemical sensor-based system for continuous air pollution monitoring for asthma patients. Besides the gas sensors, the system includes an environmental sensor for humidity and temperature measurement. The system can record the NO_2_ gas concentration as low as 10 ppb and the ozone concentration at 15 ppb with a one-minute time resolution. The sensors’ measurements are processed using a Raspberry Pi single-board computer. The resulting data are transferred to the cloud using a Wi-Fi communication module and Amazon Web Service. 

Siemuri et al. [15] presented an implementation of a wireless CAN automation module for communicating with a smart NOx sensor in a diesel engine and the engine control unit. A smart NOx gas sensor and analyzer MCS100E (SICK AG., Waldkirch, Germany) have been used with Xbee-CAN module. The work contains two Arduino-based receiver and transmitter wireless Xbee-CAN units, using ZigBee (IEEE 802.15.4) technology for data transmission. The acquired gas data from a testing W4L20 diesel engine (Wärtsilä Corporation, Helsinki, Finland) were monitored using the MATLAB-based Speedgoat monitoring tool (Speedgoat GmbH, Liebefeld, Switzerland). 

Hussain et al. [16] proposed a cloud-based low-cost air quality monitoring system. The system included several types of sensors such as gas, humidity, temperature, noise, and light sensors. Several electrochemical gas sensors were used for air quality measurements. The sensor data were processed using an ATMega32 microcontroller (Microchip Technology Inc., Chandler, AZ, USA). All the required data can be transferred to the IoT cloud using the GSM communication module. The design takes into consideration the outdoor requirements. A power management unit is responsible for both running the system and charging the system battery from solar cells. The system has the limitation of a maximum sample rate of 48 samples per day, which is changed based on the air quality variation. 

Glass et al. [17] proposed a novel low-cost sensor node for CO, NO_2_, and particulate matter air pollutant concentrations. The system used electrochemical sensor technologies for gas detection and an optical infrared sensor to measure particulate matter levels, and it used a MEMS environmental sensor for temperature and relative humidity measurements. The collected data were processed using an STM32L082 microcontroller (STMicroelectronics, Geneva, Switzerland) which was embedded in a Murata CMWX1ZZABZ-078 chipset (Murata Investment Co., Ltd., Shanghai, China). The sensor node had LoRa and Wi-Fi wireless communication modules. The sensor data were transferred to the microcontroller via five ADC channels and an I2C bus. They were further transported from the sensor node to a network server and then to an application server. The sensor node could be powered from a direct source and also recharged using solar cells. All the used sensors need to be calibrated for more accurate measurements. 

Dong et al. [18] proposed a fully integrated photoacoustic nitrogen dioxide (NO_2_) gas sensor for low-level gas concentration measurement. A 3D printing technology was used to manufacture an embedded photoacoustic cell of the NO_2_ sensor with a sensor size of 12 cm × 6.5 cm × 0.35 cm. The PLTB-450BA (OSRAM GmbH, Munich, Germany) 450 nm blue laser diode was used for NO_2_ gas detection based on the interaction of light with the gas molecules (gas spectroscopy). The acquired data were amplified and filtered with a four-stage band-pass filter for noise reduction and then converted using an analog-to-digital converter (ADC). The sensor used a microcontroller with a floating processing unit for data processing and an SPI communication protocol for communicating with other devices. The sensor was tested in several polluted areas. The results compare a NO–NO_2_–NOX gas analyzer for the same locations and show the good practical potential of the sensor. 

In this paper, the integration, testing, and evaluation of the new SGP41 (Sensirion AG, Stafa, Switzerland) metal oxide semiconductor (MOX) gas sensor are described. The sensor is a replacement for the old versions (SGP30, SGP40) produced by the same manufacturer [13,19]. The sensor has been integrated into an IoT-based portable sensor node for ambient monitoring as well as detecting and reporting the leakages of hazardous gases and vapors. The integration of this sensor improves the sensor node sensitivity as well as selectivity by adding the ability to detect the leakages of nitrogen oxides (NO, NO_2_, NOx) and the VOC-Index, which are among the main air pollutants.

The objective of this work includes testing the performance of the commercial sensor and determining whether it is suitable for integration into sensor nodes and air quality monitoring devices in locations where the leakage of harmful gases and chemical fumes is likely. The work included extensive tests to detect the leakage of nitrogen oxides in different sizes and from different sources. The sensor was also tested with a range of VOCs at different concentrations and at different test distances. This work gives a clear picture of the sensor’s performance and the speed of its response to cases of leakage, as well as the extent of time required to recover the sensor film in the sensor. The paper is structured as follows: In Section 2, we concentrate on the sensor node architecture and the details of the sensor node elements, units, and components; in addition, we discuss the specifications, parameters, and technical data regarding the selection and use of the SGP41 gas sensor. In Section 3, the testing procedure, scenarios, and testing environmental parameters are explained in detail. In Section 4, the acquired testing results for the target sensor parameters are discussed. In Section 5, the conclusions of the study are presented.

## 2. Materials and Methods

The main goal of the presented work is to update and enhance the performance of a multitask flexible sensor node and especially the gas sensor layer. The sensor node was at the Center for Life Science Automation (celisca), University of Rostock, Germany. The first version of the sensor node included 4 layers; the main one was the sensor layer with the BME688 (Bosch Sensortec, Reutlingen, Germany) and the SGP30 gas sensors [19]. In this work, the sensor node has been updated with the SGP41 gas sensor, which can detect the NOx levels as well as the VOC levels. The main layer of the sensor node will be explained in the following subsections.

### 2.1. Communication Layer

The communication layer of the sensor node is responsible for receiving the pre-processed sensor data and sending them wirelessly to the cloud server. Two types of wireless communication modules were used to achieve this goal: the ESP-WROOM-02D (Espressif Systems, Shanghai, China) Wi-Fi module, which is responsible for sending the sensor data to the cloud server via Wi-Fi protocol, and the Aconno Bluetooth module ACN52840 (Aconno, Düsseldorf, Germany), which is responsible for detecting location information using special Bluetooth-based beacon devices and then sending the location information to the cloud using the Wi-Fi module. The communication layer communicates with the processing layer for data transportation using the SPI bus. The received data were stored in 512 KB ferroelectric RAM at the communication layer to avoid data loss in case of wireless communication problems. Figure 1 shows the used communication layer.

### 2.2. Sensing Layer 

The sensing layer is responsible for collecting the required ambient data using mainly three small sensors. The sensor layer can be extended by adding a new sensor using the predefined communication buses (I2C, SPI, UART). The sensor node is designed to be flexible to use different sensing layers based on the target parameters. In the proposed work, a sensing layer with a BME688 environmental sensor, an SGP41 gas sensor, and an MS5803-05BA pressure sensor (TE connectivity, Schaffhausen, Switzerland) is used (see Figure 2).

#### SGP41 Gas Sensor

The SGP41 is a sophisticated MOX-based air quality sensor that provides two important factors regarding the levels of nitrogen oxide gases and VOC vapors in the form of NOx-Index and VOC-Index. Both indexes are unitless and have the same ranges, which start at zero for good air quality and are saturated at 500 for the worst air quality. The baseline for the NOx-Index is always 1, whereas the baseline for the VOC-Index is around 100. The SGP41 communicates with the host processor using the I2C communication protocol. The sensor has a small size of 2.44 mm × 2.44 mm × 0.85 mm and has a low power consumption of 3.0 mA at 3.3 V. It has two sensing micro hotplates (sensing elements), providing one for VOC and one for NOx gas measurements. The principle of operation of this MOX sensor is based on changing the resistance of the surface of the heated metal oxide with a change in the percentage of oxygen on the surface of the sensor, which is affected by the presence of oxidizing gases such as nitrogen oxides that increase the resistance of the surface of the sensor, or the presence of reducing gases represented by volatile organic compounds that reduce the resistance of the surface of the sensor. One of the most important characteristics that prompted us to choose this gas sensor is the built-in temperature change and relative humidity calibration algorithm, which maintains the stability of the sensor performance in the event of any unexpected change in temperature, relative humidity, or both, which is an important characteristic that affects the performance of gas sensors and enables unique long-term stability as well as low drift [20,21,22,23,24].

### 2.3. Processing Layer

The processing layer is the central layer of the sensor node. It is responsible for sampling, processing, and transferring all sensor data. The main element of the processing node is the NXP MKL27Z256LH4 microcontroller (NXP Semiconductors, Eindhoven, Netherlands), which is a 32-bit ARM microcontroller with a processing clock of 32 MHz. It has all the required buses, such as USB Type-C, I2C, SPI, and UART interface, to communicate with different sensors and modules in the sensor node layers. The USB Type-C bus can be used for powering the sensor node, charging the battery for mobile applications, and system configuration. The processing layer has the MCP79411 battery-backed I2C real-time clock (Microchip Technology Inc., Arizona, USA) for generating a time stamp for sensing layer measurements. Figure 3 shows the front and back sides of the processing layer.

### 2.4. Power Management Layer

The power layer is an optional layer for providing the power source to the sensor node in case a typical USB Type-C interface is not available in the hosted environments. Two power input interfaces can be provided by the power layer. A 24 V direct DC supply is used for implementation with some types of humanoid and tank robots. The second power interface is for a typical rechargeable battery (3.7 V) which is widely used for mobile applications. The rechargeable battery attached to the second power layer interface can be charged from the first 24 V DC supply interface as well as from the USB Type-C interface of the processing layer board. Figure 4 shows the power layer of the proposed sensor node.

## 3. System Testing 

The SGP41 sensor was tested for its two NOx-Index and VOC-Index parameters with different gases and VOCs. Several testing procedures, environments, sample volumes, and distances from the sensor node were used to evaluate the sensor performance and also to investigate the best condition for using the sensor node in stationary as well as in mobile ambient monitoring applications. All tests were carried out in the laboratories of the Center for Life Sciences Automation (celisca), University of Rostock, Germany, which are equipped with a programmed air ventilation system to keep the temperature within 22 ± 0.5 degrees Celsius and the relative humidity around 50.0 ± 2.0%. In addition, both SGP41 and BME688 gas sensors have calibration algorithms to avoid any sensor drift caused by the change in temperature and relative humidity. The following two subsections explain the testing procedures and results for NOx-Index and VOC-Index. 

### 3.1. NOx-Index Testing

The NOx-Index refers to the levels of nitrogen oxides in indoor air. This index changes with increasing concentrations of NO, NO_2_, or both (NOx) in the tested air sample. For this purpose, special SKC gas sample pages (SKC, Inc., Eighty Four, PA, USA) of different sizes (1 L, 10 L) and a small electrical pump KNF with a max pressure of 2.4 bar (KNF Neuberger GmbH, Freiburg, Germany) were used for collecting the NO, NO_2_, and NOx gas mixture from different sources such as a pure NO_2_ gas bottle, benzene fuel engine exhaust, and diesel fuel engine exhaust. The testing of the nitrogen oxide gases was implemented using a special gas testing chamber designed for this purpose and made of polyvinyl chloride (PVC) (see Figure 5). It has dimensions of 52 cm in height, 42 cm in length, and 30 cm in width with a total volume of 65 L. The gas testing chamber has a small fan to uniformly mix gas samples with the air inside the chamber to ensure accurate sensor response. In addition, the fan is used to evacuate the gas sample from the gas chamber after the testing phase and replace it with fresh air for the next testing phase (the fan continuously works until the sensor measures normal NOx-Index baseline).

In each test, we collected the gas sample in a 10 L polyvinyl fluoride gas sample bag. The required test volume of the gas was taken using the small pump. For volumes less than 1 L, a 60 mL PVC syringe was used (see Figure 6). The tested volumes were 0.2 L, 0.5 L, 0.7 L, 1 L, 2 L, 3 L, 4 L, 5 L, and 6 L. Before testing, the SGP41 sensor was warmed up for 5 min. The baseline for NOx-Index should be 1. Any other value more or less indicates a NOx measurement or a calibration error. After the required volume was collected, the gas sample was discharged into the gas chamber through a small hole for the gas inlet in the upper side of the chamber. The gas sample was mixed with air in the chamber by using the fan, which was turned on for a minimum of two minutes. A specific C++-based program was written to collect the sensor data. Based on the Nyquist theorem, we used a 2 Hz measurement frequency to obtain two samples per second from the SGP41 sensor, which provides one sample per second. The software allows the recording of the sensor node measurements for all sensors and storing them as a text table for a 15 min time window. Furthermore, the sensor node sends the recorded data directly via the Wi-Fi module to the cloud, which stores all the acquired data in the main institute database. 

NO_2_ gas with a 200 ppm concentration was used in sensor node testing, while two samples of NOx gases were taken from the exhausts of two different cars with different engine fuels. Figure 7, Figure 8 and Figure 9 show the SGP41 sensor response for NO_2_, NOx (first sample), and NOx (second sample), respectively. 

### 3.2. VOC-Index Testing

The VOC-Index represents the concentration of the VOCs in the air at five levels from good to hazardous air quality. The SGP41 sensor testing for the VOC-Index was implemented in parallel with the IAQ-Index of the BME688 sensor, which was included in the same sensing layer. Both indexes refer to the level of air quality range from 0 to 500 representing the best to the worst quality range, respectively. The testing was implemented inside a Secuflow fume sample preparation hood (Waldner Holding GmbH & Co. KG, Wangen im Allgäu, Germany) that was designed for chemical sample preparation in laboratories. The SGP41 sensor was tested with 10 different VOC solvents, namely acetone, acetonitrile, benzene, diethyl ether, ethanol, formic acid, hexane, isopropanol, methanol, and toluene. Four volumes (5 µL, 10 µL, 50 µL, and 100 µL) from each VOC were tested. The VOC samples were taken using accurate Eppendorf pipettes (Eppendorf SE, Hamburg, Germany) and injected into a 15 cm diameter Petri dish placed directly under the sensor node. The sensor node was fixed on an adjustable height stand. Two distances (40 cm and 100 cm) between the sensor node and the VOC source were used for system testing. Figure 10 shows the testing hood with used adjustable stand. Figure 11, Figure 12, Figure 13, Figure 14, Figure 15, Figure 16, Figure 17, Figure 18, Figure 19 and Figure 20 show the sensor node test results of the VOC-Index of the SGP41 gas sensor and the IAQ-Index of the BME688 sensor for the selected VOC solvents.

The step response time (T90) and the recovery time for the SGP41 NOx-Index are presented in Figure 21a,b, respectively. The step response time represents the time required for the sensor to reach 90% of its final output value for an abrupt change to the measured NOx samples. In contrast, the recovery time represents the time required to let the sensor return from the maximum recorded value to the baseline value of the testing environment.

The step response time (T90) and the recovery time for the SGP41 VOC-Index and the BME688 IAQ-Index for several selected VOC solvents with the same volume (100 µL) with a fixed distance between the sensor node and the VOC source of 40 cm are presented in Figure 22a,b, respectively.

## 4. Discussion

The NOx-Index testing results show that the SGP41 sensor can detect a small NO_2_ gas leakage starting from 0.3% (0.2 L/65 L volume) in indoor environments. The baseline for the NOx-Index is 1 in a range that extends to 500, and this baseline can be used as a stable threshold for abnormal NOx gas concentration notification, whereas a higher threshold (NOx-Index > 20) can be used for triggering alarms. The SGP41 sensor testing for the car exhaust sample of a four-cylinder benzene engine (sample 1) shows that the SGP41 sensor cannot detect a low concentration of 0.3% (0.2 L/65 L volume) and the sensor response starts with 0.77% (0.5 L/65 L volume) which results in NOx-Index = 4. The SGP41 sensor response to the second car sample of four-cylinder diesel engine exhaust revealed that it does not respond to the NOx mixture volumes of less than 0.7% (0.5 L/65 L volume) with NOx-Index = 4. 

The VOC-Index (SGP41 sensor) and IAQ-Index (BME688 sensor) testing revealed that both sensors have a good response to all tested solvent volumes. The response of the SGP41 sensor is better than that of the BME688 for 9/10 of the tested VOCs, whereas the BME688 shows a better response to formic acid. As one of the purposes of the system design is to alarm the workers in the target environment in case of any dangerous levels of toxic and hazardous gases, a threshold should be selected for triggering the alarm. In the presented work, a 200 VOC-Index and IAQ-Index can be used for this purpose. Based on that, the sensor response for the two used test distances is summarized in Table 1. The results in Table 1 revealed that the SGP41 sensor can detect all the VOC solvents ≥ 100 µL from both test distances. The BME688 failed to detect benzene, diethyl ether, hexane, and toluene from a 100 cm distance and failed to detect the alarm threshold for hexane and toluene from a 40 cm distance. These results are very encouraging for the adoption of SGP41 as one of the main gas sensors in our sensor node. Figure 23 shows the maximum readings for each material with different concentrations for both IAQ-Index and VOC-Index (a) from 40 cm and (b) from 100 cm. 

The step response time for the SGP41 sensor for NOx-Index shows that the maximum recorded T90 for the selected samples with two volumes of 1 L and 6 L was 260.1 s for a 1 L of NOx gas mixture sample produced by a benzene engine and the minimum recorded T90 step time response was 113.4 for a 1 L of NOx gas mixture produced by a diesel engine. When comparing the results in Figure 21, we find that the response and recovery times vary at some volumes according to different samples, and this can also be distinguished by referring to Figure 7, Figure 8 and Figure 9, in which some abnormal readings appear that may be due to the SGP41 sensor repeatability [24].

This range of NOx sensor T90 is still useful because for our application we will use specific NOx-Index thresholds, for example, NOx-Index = 5 which requires less time (4 s for NO_2_ 9% volume sample). Furthermore, the maximum recorded recovery time for the SGP41 NOx-Index is 477.5 s for the NO_2_ (200 ppm) sample, and the minimum recorded recovery time is 271 for the NOx diesel engine sample. The recovery time is a very important factor for our application since it determines the intervals at which measurements can be meaningfully carried out. 

The step time response for the SGP41 sensor for VOC-Index shows that the maximum recorded T90 time is 268.2 s for 100 µL of acetonitrile while the minimum recorded T90 step time response is 96.3 s for 100 µL of benzene. The step time response for the IAQ-Index of the BME688 sensor behaves differently with the selected/tested VOCs, where the maximum recorded T90 is 390.6 for 100 µL of benzene while the minimum recorded T90 response time is 27 s for 100 µL of formic acid. Furthermore, the maximum recorded recovery time for the SGP41 VOC-Index is 1210 s for a 100 µL of methanol sample from a 40 cm distance and the minimum recorded recovery time is 273 for ethanol with the same volume and distance. Moreover, the maximum recorded recovery time for the BME688 IAQ-Index is 1300 s for a toluene sample, and the minimum recorded recovery time is 353 for benzene for a 100 µL volume and 40 cm distance. All recorded response and recovery times are functional for the work, where the alarm indication can be activated during the recovery time after the first confirmed danger level measurement. 

Comparing the current results of the general performance of the SGP41 gas sensor with the results in previous studies [11,13,19] for SGP30, SGP40, and BME680 shows that the SGP41 sensor has a better response to most of the VOCs that were previously tested, in addition to its better selectivity due to it containing an additional sensing layer for nitrogen oxides, which makes it a better choice for new designs of air pollution monitoring devices and harmful gas detectors. 

## 5. Conclusions

In this work, the integration and testing of the novel SGP41 gas sensor in a portable sensor node for ambient monitoring are presented. The sensor can measure the index levels for both NOx gas mixtures and VOCs, which are the main parameters for air pollution. The sensor has been combined with another gas sensor, BME688, and both of them have been tested with several VOC solvents. The testing results show that the SGP41 has a better response for 9/10 tested VOC solvents and can detect the level of NOx gases, which is a unique and important feature for hazardous and toxic gas detection and alarm systems. Based on system testing, the SGP41 sensor can detect a small NOx gas leakage starting from 0.3% (0.2 L/65 L volume) and can detect all the VOC solvents ≥ 100 μL from both test distances. The sensor performance evaluation revealed that the response and recovery time for both SGP41 and BME688 are useful for the design purpose. 

In future work, the sensor layer can be expanded with the inclusion of a novel sensor for detecting different gases and/or air pollutants such as H_2_, CO, SO_2_, O_3_, and PM2.5. 

## Figures and Tables

**Figure 1 sensors-22-08161-f001:**
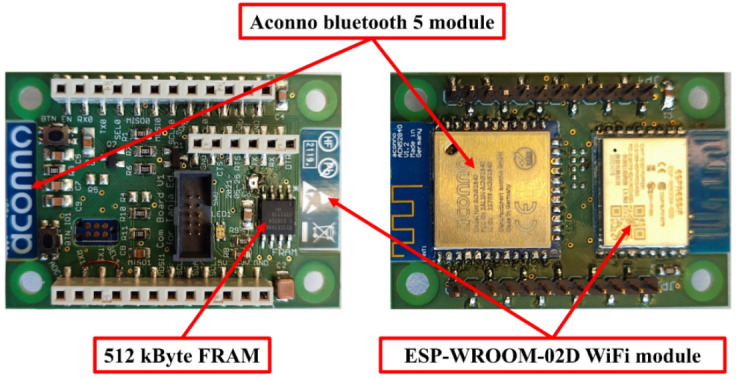
Communication layer (front and back views).

**Figure 2 sensors-22-08161-f002:**
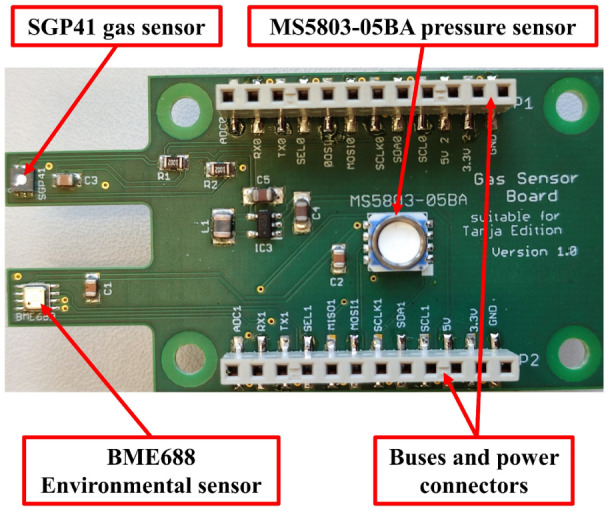
Sensing layer.

**Figure 3 sensors-22-08161-f003:**
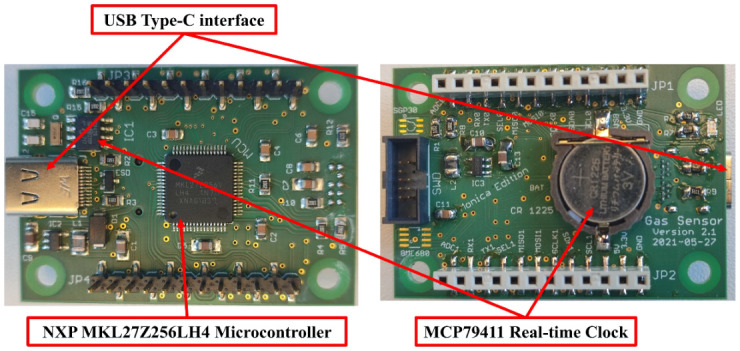
Processing layer (front and back views).

**Figure 4 sensors-22-08161-f004:**
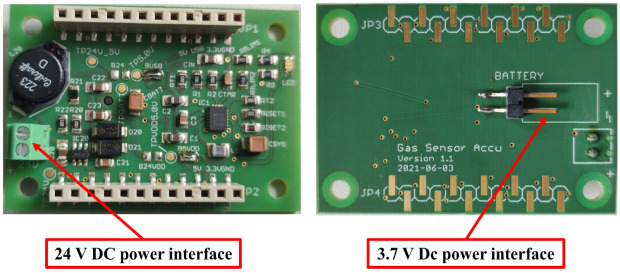
Power layer (front and back views).

**Figure 5 sensors-22-08161-f005:**
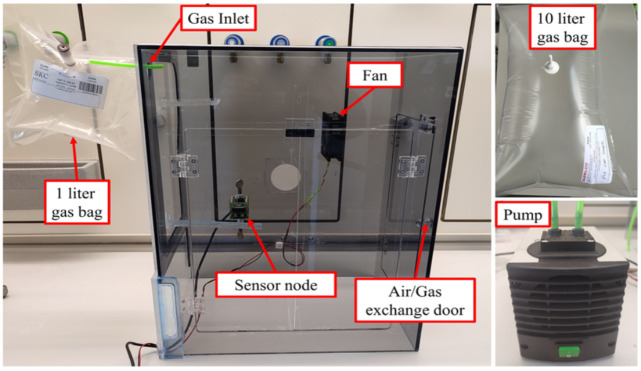
The used gas testing chamber, 10 L gas sample bag, and small pump.

**Figure 6 sensors-22-08161-f006:**
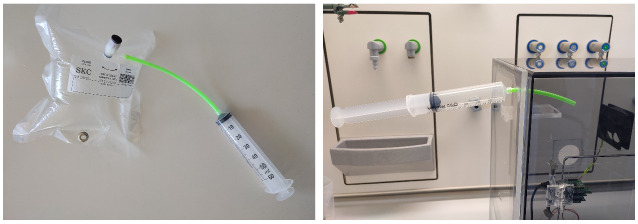
Small volume gas sample testing <1 L.

**Figure 7 sensors-22-08161-f007:**
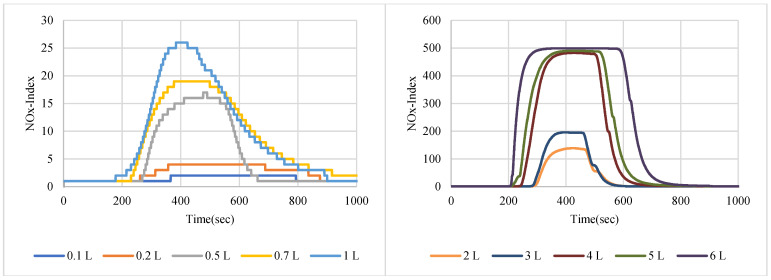
NOx-Index response for NO_2_ gas sample (200 ppm).

**Figure 8 sensors-22-08161-f008:**
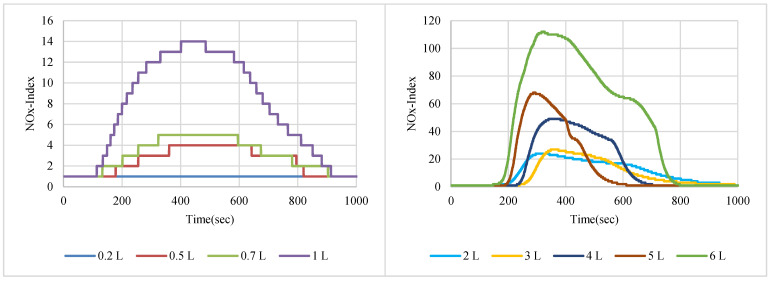
NOx-Index response for NOx car exhaust gas sample 1.

**Figure 9 sensors-22-08161-f009:**
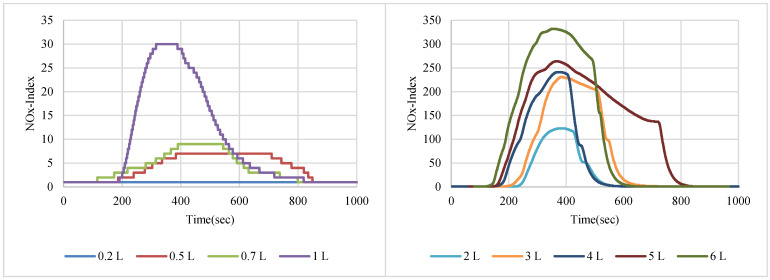
NOx-Index response for NOx car exhaust gas sample 2.

**Figure 10 sensors-22-08161-f010:**
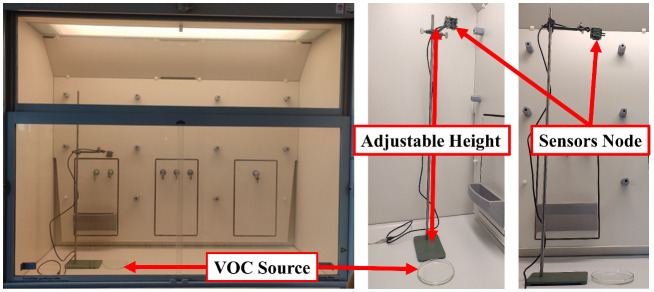
The testing hood with adjustable height stand.

**Figure 11 sensors-22-08161-f011:**
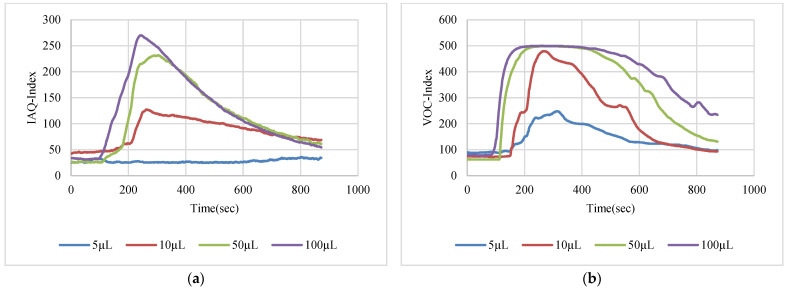
Test results for acetone: (**a**) BME688 40 cm, (**b**) SGP41 40 cm, (**c**) BME688 100 cm, (**d**) SGP41 100 cm.

**Figure 12 sensors-22-08161-f012:**
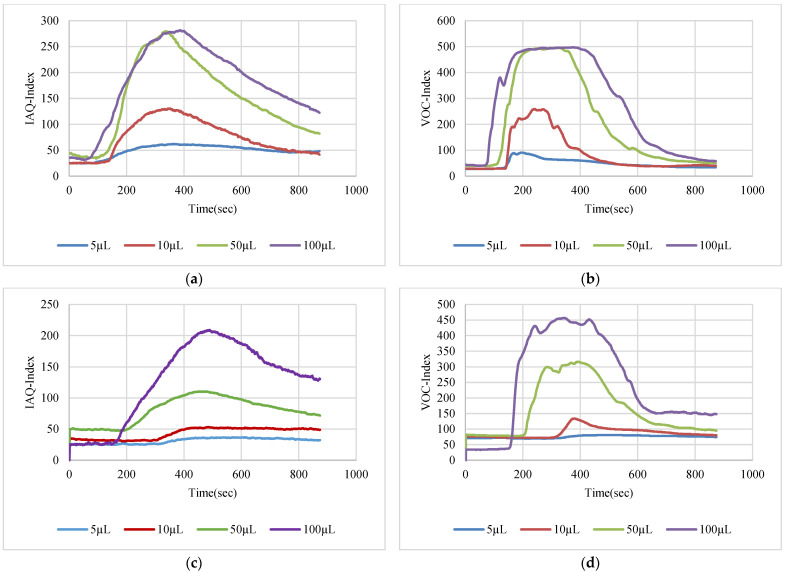
Test results for acetonitrile: (**a**) BME688 40 cm, (**b**) SGP41 40 cm, (**c**) BME688 100 cm, (**d**) SGP41 100 cm.

**Figure 13 sensors-22-08161-f013:**
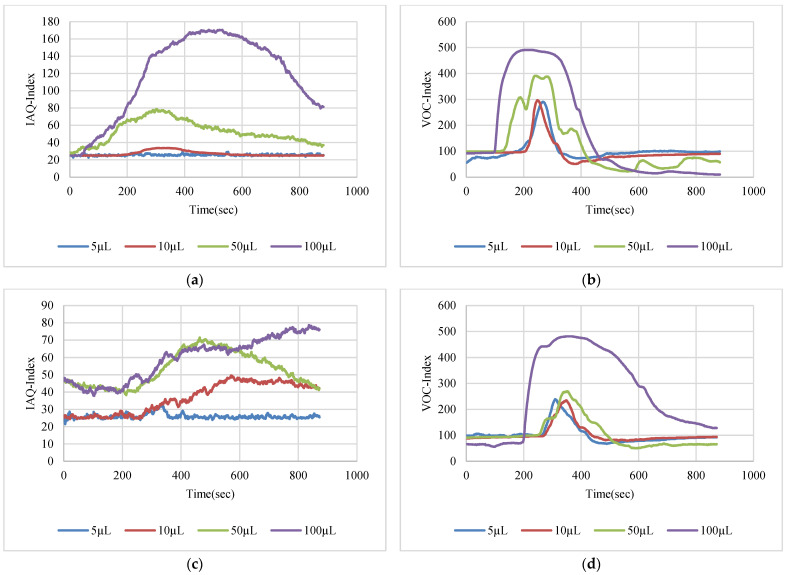
Test results for benzene: (**a**) BME688 40 cm, (**b**) SGP41 40 cm, (**c**) BME688 100 cm, (**d**) SGP41 100 cm.

**Figure 14 sensors-22-08161-f014:**
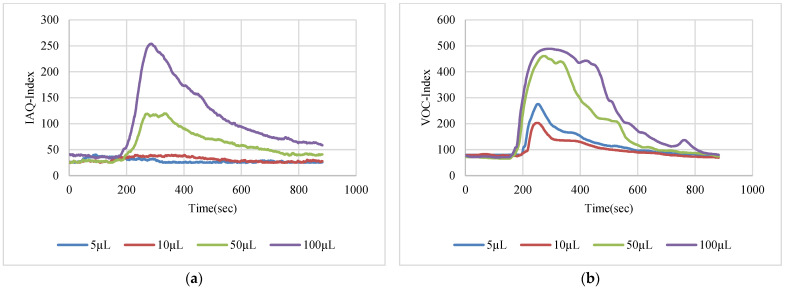
Test results for diethyl ether: (**a**) BME688 40 cm, (**b**) SGP41 40 cm, (**c**) BME688 100 cm, (**d**) SGP41 100 cm.

**Figure 15 sensors-22-08161-f015:**
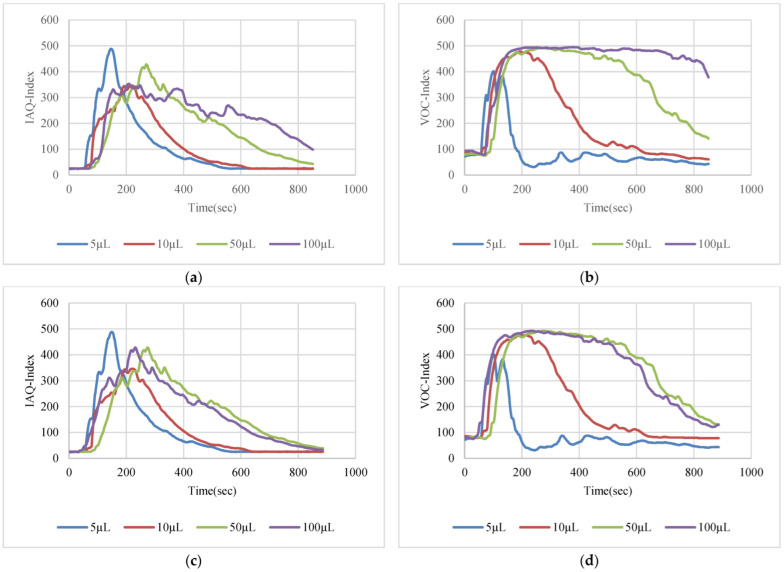
Test results for ethanol: (**a**) BME688 40 cm, (**b**) SGP41 40 cm, (**c**) BME688 100 cm, (**d**) SGP41 100 cm.

**Figure 16 sensors-22-08161-f016:**
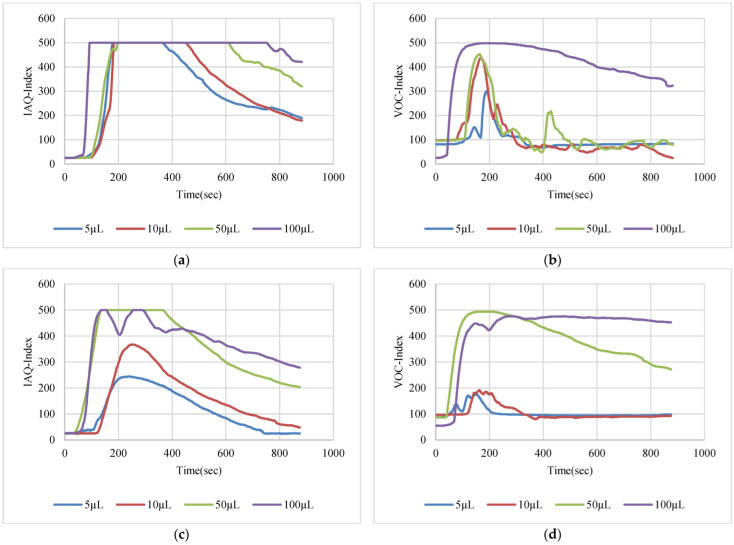
Test results for formic acid: (**a**) BME688 40 cm, (**b**) SGP41 40 cm, (**c**) BME688 100 cm, (**d**) SGP41 100 cm.

**Figure 17 sensors-22-08161-f017:**
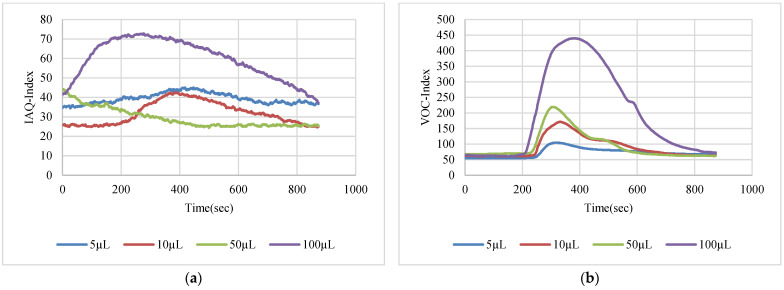
Test results for hexane: (**a**) BME688 40 cm, (**b**) SGP41 40 cm, (**c**) BME688 100 cm, (**d**) SGP41 100 cm.

**Figure 18 sensors-22-08161-f018:**
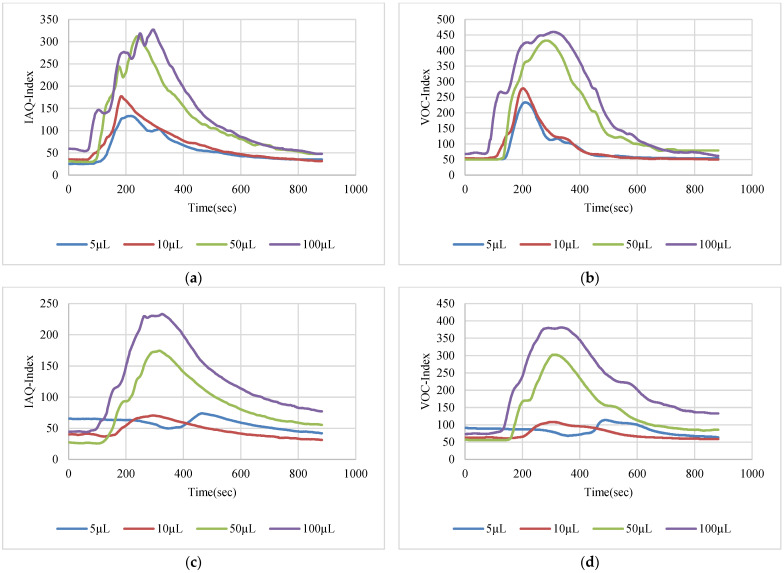
Test results for isopropanol: (**a**) BME688 40 cm, (**b**) SGP41 40 cm, (**c**) BME688 100 cm, (**d**) SGP41 100 cm.

**Figure 19 sensors-22-08161-f019:**
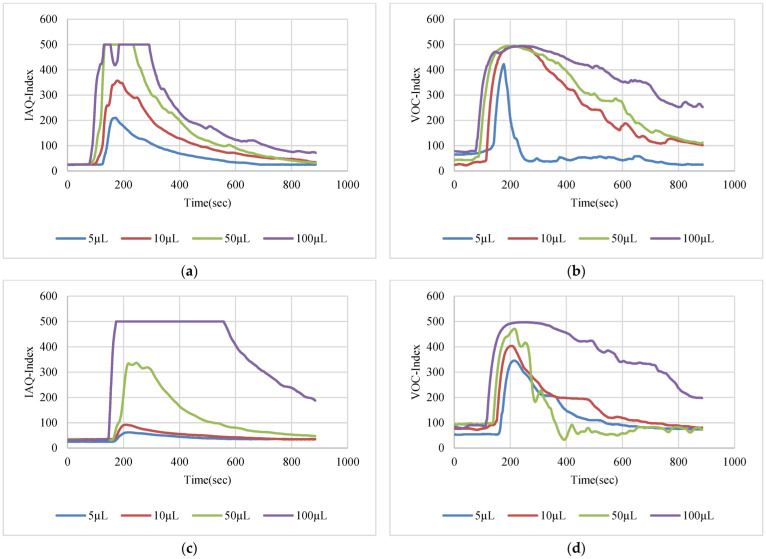
Test results for methanol: (**a**) BME688 40 cm, (**b**) SGP41 40 cm, (**c**) BME688 100 cm, (**d**) SGP41 100 cm.

**Figure 20 sensors-22-08161-f020:**
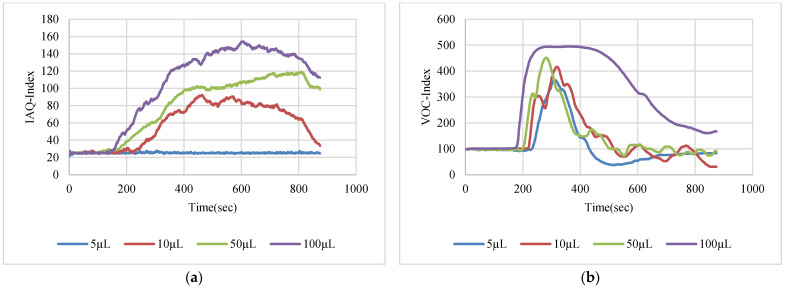
Test results for toluene: (**a**) BME688 40 cm, (**b**) SGP41 40 cm, (**c**) BME688 100 cm, (**d**) SGP41 100 cm.

**Figure 21 sensors-22-08161-f021:**
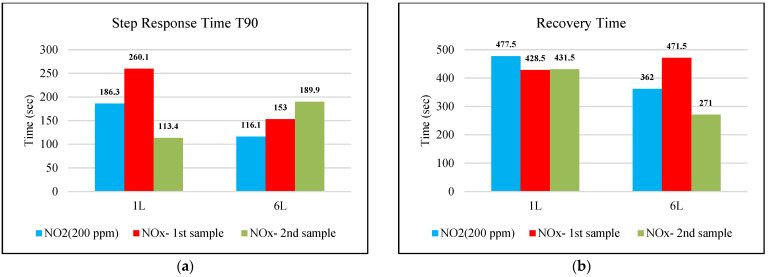
(**a**) SGP41 step response time T90 for NOx gases, (**b**) SGP41 recovery time for NOx gases.

**Figure 22 sensors-22-08161-f022:**
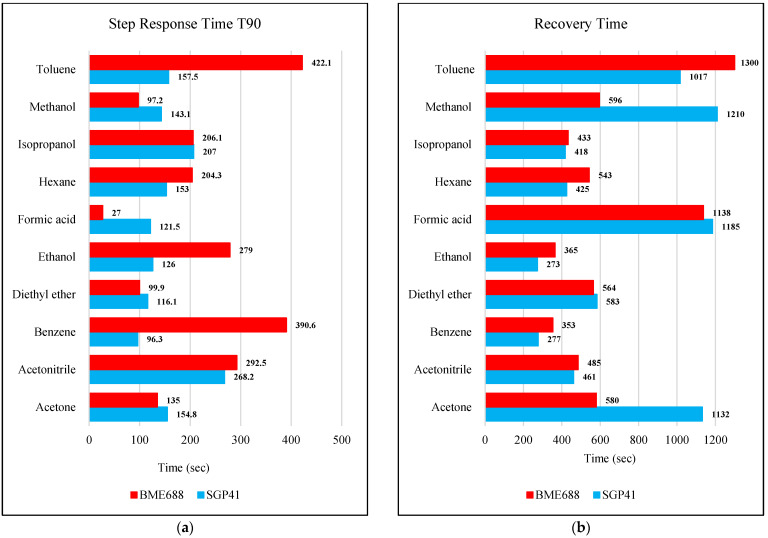
(**a**) SGP41 VOC-Index and BME688 IAQ-Index step response time T90 for selected VOCs of 100 µL volume from a 40 cm distance between the sensors and the leakage source, (**b**) SGP41 VOC-Index and BME688 IAQ-Index recovery time for selected VOCs of 100 µL volume from 40 cm distance between the sensors and the leakage source.

**Figure 23 sensors-22-08161-f023:**
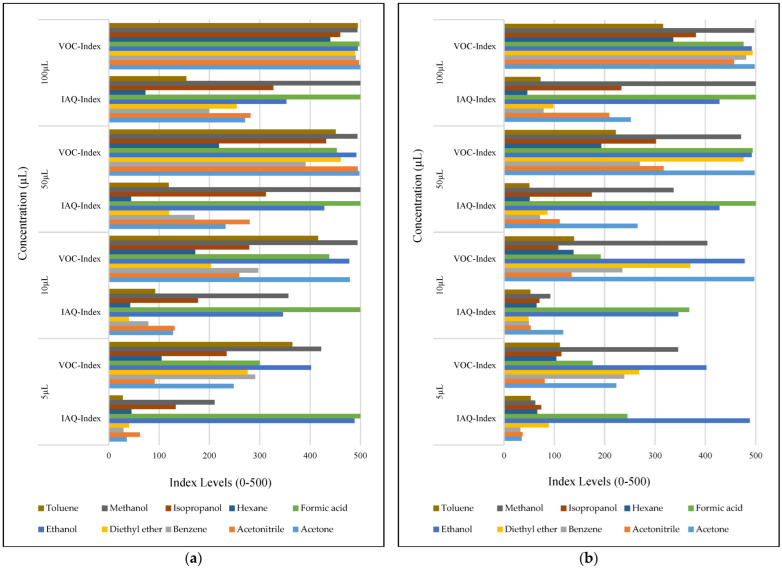
Maximum sensor responses for the tested VOCs from (**a**) 40 cm distance, (**b**) 100 cm distance.

**Table 1 sensors-22-08161-t001:** SGP41 VOC-Index and BME688 IAQ-Index success responses to 200 index thresholds for the tested VOCs.

VOC	SGP41 (40 cm)	SGP41 (100 cm)	BME688 (40 cm)	BME688 (100 cm)
Acetone	≥5 μL	≥5 μL	≥50 μL	≥50 μL
Acetonitrile	≥10 μL	≥50 μL	≥50 μL	≥100 μL
Benzene	≥5 μL	≥5 μL	≥50 μL	-
Diethyl ether	≥5 μL	≥5 μL	≥100 μL	-
Ethanol	≥5 μL	≥5 μL	≥5 μL	≥5 μL
Formic acid	≥5 μL	≥50 μL	≥5 μL	≥5 μL
Hexane	≥50 μL	≥100 μL	-	-
Isopropanol	≥5 μL	≥50 μL	≥50 μL	≥100 μL
Methanol	≥5 μL	≥5 μL	≥5 μL	≥5 μL
Toluene	≥5 μL	≥50 μL	-	-

## Data Availability

The data presented in this work are available on request from the corresponding author.

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
