# Peer review of "Evaluating a Novel Gas Sensor for Ambient Monitoring in Automated Life Science Laboratories"

_sensors, 2022, doi:10.3390/s22218161_

Round 1

Reviewer 1 Report

In this manuscript, the authors used a commercial MOX gas sensor to integrate into an IoT-based mobile sensor node for detecting gases and vapors. They have claimed that the presented system improves sensitivity and selectivity by adding the ability to detect the leakages of nitrogen oxides. The article provides sufficient data, but the scientific review has received less attention. Review articles (like DOI:10.3390/s100302088) can help you. Also, the following points should be considered in the manuscript.

1-     Abstract: The authors wrote: “novel MOX gas sensors have been embedded…”. Replace it with “a commercial XXX MOX gas sensor ….

2-     Abstract: The abstract should begin with a brief and precise statement of the problem, followed by a description of the study method and design, the major results, and the conclusions reached. But in this manuscript, the abstract is presented in a general mode.

3-     Abstract: The exact value of sensor performance improvement should be mentioned in the abstract.

4-      The aim of the manuscript should be presented with more details at the end of the introduction.

5-      The background literature review is very brief in the introduction. Show the importance of the work by adding appropriate and related references.

6-      I strongly recommend comparing new metal oxide sensors for VOC and NOx sensing with the gas sensor presented in the introduction. These articles are some examples: DOI:10.3390/s21051826, DOI: 10.1088/1361-6528/abfd54 and …)

7-      The standard form of reference number is not used.

8-      Give a detailed description of the SGP41 metal oxide sensor (material, how it works, etc.).

9-      Why did the authors use this commercial sensor? Does this sensor have a special and unique feature? Or is it part of your group's business plan?

10-  Gas testing Experimental: the name of materials and instruments should be presented with details.

11-   What was the temperature and humidity used for the testing study in Fig 7, 8, 9, 11, 12 and 13? Why you choose that temperature? the reader will expect to have a discussion on the influence of both factors simultaneously on the samples. How they will actually affect each other. Do you have the same response curve for each testing under a certain temperature and humidity?  It is important to have the same response and recovery shape each time you tested for under different conditions. This meant that there was no serious sensing poisoning to your samples and they were full recoverable to the initial stage. You can put them in the supplementary data. Or maybe you can refer to the reference.

12-   When you introduced VOC, did you mix it with purified air or N2? To emulate the real sensing environment, it has to be air.

13-   When people talk about the sensor, we care about 3Ss. Sensitivity, Stability and Selectivity.  Do you have any discussion about the stability? This information may be provided by the manufacturer.

14-   Reduce the number of your group's articles in ref [13-16] in the last paragraph introduction. “In this paper, the integration, testing, and evaluation of the new SGP41 (Sensirion AG, Stafa, Switzerland) metal oxide semiconductor (MOX) gas sensor is described. The sensor is a replacement for the old versions (SGP30, SGP40) of the same manufacturer [13]–[16].”

Author Response

Dear reviewer, thank you very much for your efforts and valuable suggestion to enhance our manuscript, we try to follow and modify our paper based on all the suggestions in the comments as below:
General concept comments
- The Abstract, introduction, materials and methods and references sections have been updated.
Specific comments
1- In Abstract line 16. The paragraph has been updated to “commercial MOX gas sensor “.
2- In the Abstract lines, 15-16 and 19-28 have been added and the abstract updated based on comment no. 2.
3- In Abstract lines 27-28, the abstract is updated based on comment no. 3.
4- The aim of the manuscript was presented in more detail at the end of the introduction in lines 116-123.
5- The introduction has been updated in lines 45-53, and 112-120.
6- This point has been explained in lines 184-190.
7- We use Zotero software for automatic references which is recommended by the Sensors journal.
8- A detailed description of the SGP41 metal oxide sensor has been included in lines 184-196.
9- The reason behind selecting this sensor has been mentioned in lines 180-181, and 191-196.
10- The name of materials and instruments have been updated in lines 243-245 and lines 301-302.
11- The temperature and humidity setting details have been explained in lines 237-243.
12- The samples have been tested in fresh purified air.
13- The Stability information has been updated in lines 190-198
14- The references to the group's articles have been reduced in line 107

Reviewer 2 Report

In this paper, the authors deployed MOX gas sensors to an IoT environmental sensor node for detection and warning of hazardous gases. They examined various parameters such as NOx-Index and VOC-index for the sensors under different conditions. Sensor performances yielded beneficial results for both response and recovery times. However, the manuscript needs to be improved in various aspects as described below.

1)      “Volatile organic compounds” (VOC) is given in the abstract. In later sections (lines 28, 32, 154, and 258), using VOC abridgement is sufficient.

2)      Does the sensor have selectivity to gases other than NOx and VOC? Has it been tested?

3)      What is the reason for the differences in the measurements (1L and 6L) in Figure 21? It should be discussed depending on the literature.

4)      In Figure 22 and Table 1, the comparison of VOC-index for two sensors is given very well. Only the word "Toluene/Toluol" in the grap and table were misspelled.

5)      In the discussion section, the evaluations and comments should be supported and discussed with up-to-date citations.

6)      In the conclusion section, the results obtained should be presented in more detail.

Author Response

Dear reviewer, thank you very much for your efforts and valuable suggestion to enhance our manuscript, we try to follow and modify our paper based on all the suggestions in the comments as below:
Answers to comments
1. The mentioned “Volatile organic compounds” were replaced with VOC abridgment.
2. The sensor tested only with NOx and VOC.
3. The answer has been included in lines 443-447.
4. The word “Toluol” has been replaced with “Toluene” in the whole manuscript
5. The comment has been processed in lines 471-477
6. In the conclusion section, the results are explained in more detail in lines 480-482.

Round 2

Reviewer 1 Report

The new version of the manuscript is satisfyingly improved.

- The reference format should be modified.